# Coupling coordination between environment, economy and tourism: A case study of China

**Zhizhu Lai[1], Dongmei Ge[2], Haibin Xia[1], Yanlin Yue[1], Zheng Wang[1,3]\***

**1** Key Laboratory of Geographic Information Science, Ministry of Education, East China Normal University, Shanghai, China, **2** School of Ecological Engineering, Guizhou University of Engineering Science, Bijie, Guizhou, China, **3** Institute of Science and Development, Chinese Academy of Sciences, Beijing, China

\* wangzheng@mail.casipm.ac.cn

**Data Availability Statement:** All relevant data are within the manuscript and its Supporting Information files.

**Funding:** This research was supported by National Natural Science Foundation of China (41671396),

## Abstract

How to achieve the sustainable and coordinated development of the ecological environment, the economy and tourism has already received much attention. In this paper, a comprehensive evaluation index system of the ecological environment, the economy and tourism is established, and the coupling degrees and coordination degrees of the three subsystems of 31 provinces in China from 2003 to 2017 are calculated. The results show that the average coupling degree and average coordination degree have fluctuating upward trends during the period of 2003–2017. According to the spatial distribution of the coupling degrees and coordination degrees, the coastal provinces and Chongqing, with their high ecological environment pressure and good economic development, have low coupling and extremely high coordination levels. The vast central and western provinces with good ecological environment protection and economic backwardness have high coupling and low coordination development level. From the perspective of coordinated development, only seven of the western provinces and three middle-eastern provinces possess basic coordinated development of the ecological environment, the economy and tourism. The remaining 21 provinces are over-utilizing or sacrificing their ecological environments, among which eleven eastern provinces have an advanced economy or advanced tourism and five southwestern provinces with high tourism resource endowments have an advanced tourism.

## 1 Introduction

The rapid growth of China's economy has led to serious environmental problems, and the deterioration of the ecological environment has further limited economic development [1–3]. With the continuous improvement of people's living standards, tourism has become a fast-growing emerging industry, increasing from 0.49 trillion Yuan in 2003 to 5.40 trillion Yuan in 2017. The ecological environment provides the basic guarantee for the regional economy, and the environment is an important criterion for measuring the quality of a tourist area. The state of the ecological environment affects the tourism experience, and it also plays a role in limiting or boosting the speed and progress of tourism development [4]. In addition, changes in the ecological environment have strong counter-effects on tourism development, especially in

Natural Science Foundation of Shanghai (19ZR1415200) and Guizhou Science and Technology Department (JLKB[2012]23 and LH [2014]7532).

**Competing interests:** The authors have declared that no competing interests exist.

terms of climate warming [5–6]. The impacts of economic and tourism development on the environment are both advantageous and disadvantageous. Rapid economic development and tourism development will have great negative effects on the environment, and at the same time, they may guide or accelerate the improvement of the quality of the ecological environment. How to coordinate the regional economy and tourism with ecological environment while vigorously developing the regional economy and tourism has become an important issue that needs to be studied and solved.

The relationship between ecology and economy has been the research hotspot of many scholars. Many theories and models have been proposed, such as environmental Kuznets curve (EKC) [7–9], coordinated development theory [10], economic-energy-environmental impact model [11], decoupling theory [12–14] and coupling model [15]. With the rapid development of tourism, some scholars have begun to explore the relationship between tourism, the environment and the economy, such as the relationship between tourism and ecological environment [16–23], the relationship between economic development and tourism [24–31], and interactive relationship between the environment, economy and tourism [32–36]. For example, Pang et al. found that the inbound tourism in eastern China has a significant impact on regional economic growth, and the growth of the inbound tourism in the eastern region is the Granger cause for the growth of its tertiary industry [25]. Petrosillo et al. [32] and Lacitignola et al. [33] constructed a model that reflects the relationship between environmental quality and economic society. Moreover, Wei et al. used mathematical concepts and tools to study the sustainable relationship between social ecosystems and regional tourism [34]. Wu et al. studied the causal relationship between international tourism revenue and economic growth in 11 provinces in eastern China [35]. Using a combination of several quantitative methods (including parameter analysis, fuzzy classification, regression analysis and gray correlation), Lu et al. evaluated the coordination development of the ecological environment, economic growth and the tourism industry at the provincial level and prefectural level of Gansu Province in China [36].

Coupling coordination theory can be used to describe the degree of interaction between two or more subsystems. The degree of coupling can describe the intensity of the interaction, while the degree of coordination reflects the intensity of cooperative development. The coupling coordination theory not only has the ability of comprehensive evaluation system, but also has intuitiveness and easy interpretation, therefore it has been widely used in empirical applications [37–46]. In recent years, some scholars have begun to study the coordination of ecological environment, the economy and the tourism industry using coupling coordination theory. For example, Yuan et al. applied a coordinated development model to study the coordination development of the regional environment, the economy and the tourism of western Hunan Province in China [47]. Zhou et al. measured the coupling degree of the economy, the ecological environment and the tourism industry for 11 provinces in the Yangtze River Economic Belt from 2002 to 2013 [48]. These studies illustrate the importance of the coupling and coordinated relationship of the ecological environment, the economy and the tourism industry. However, these studies are limited to a single province or a national region in static time and ignore spatial-temporal evolution analysis, or they consider the spatial-temporal evolution but the study area is limited to an economic region.

In this work, we attempt to explore the following two questions: 1) What is the coordination relationship between the ecological environment, the economic development level and the tourism industry on provincial scale? 2) What is the temporal and spatial evolution of this relationship? Therefore, we first established an index system of the ecological environment, economic development level and tourism industry, and then comprehensively evaluated the ecological environment, economic development level and tourism industry of 31 provinces

in China from 2003 to 2017. Then a three-subsystem coupling coordination model is constructed. Finally, the spatial-temporal evolution analysis of the coordination relationship among three subsystems of 31 provinces in China is made.

## 2 Index systems and methods

### 2.1 Index systems

To explore the coupling relationship between the ecological environment, the economic development level and the tourism industry in China, we construct an aggregated index system to evaluate the ecological environmental, economic and tourism effects using previously developed indexes [49–50]. The indexes are primarily selected according to the following criteria [40, 51]: (1) they are the most cited indexes; (2) they cover the components of the ecological environment, economy and tourism; and (3) they have good representativeness, availability and integrity. According to the relevant literature [36, 39–41, 44, 47, 52–57] and the availability of the data, we select 27 basic-class indexes for the three subsystems (Table 1). Among the 27 basic-class indexes, except for the five indexes that reflect the pressure of the ecological environment (which are A21, A22, A23, A24 and A25) and are negative indexes (the smaller the better), the other indexes are all positive indicators (the larger the better). The sample period of this study is from 2003 to 2017, and the data are obtained from the China Statistical

**Table 1. Indexes of three subsystems.**

| Subsystem | First-class index | Weight | Basic-class index | Weight |
|---|---|---|---|---|
| A. Ecological environment | A1. Ecological environment endowment | 0.7866 | A11. Wetland area per capita (m$^2$) | 0.8190 |
| | | | A12. Forest cover rate (%) | 0.1230 |
| | | | A13. Green areas per capita (m$^2$) | 0.0580 |
| | A2. Ecological environment pressure | 0.0479 | A21. Discharged volume of industrial SO2 (tons) | 0.1864 |
| | | | A22. Discharge of smoke and dust (tons) | 0.2226 |
| | | | A23. Discharge of waste water (tons) | 0.1898 |
| | | | A24. Discharge of ammonia nitrogen from waste water per capita (tons) | 0.1898 |
| | | | A25. Discharge of COD emissions from waste water per capita (tons) | 0.2114 |
| | A3. Ecological environment response | 0.1655 | A31. Soil erosion control area per capita (m$^2$) | 0.5643 |
| | | | A32. Investment of pollution treatment per capita (Yuan) | 0.4357 |
| B. Economic development level | B1. Economic level | 0.2543 | B11. Per capita GDP (Yuan) | 0.5763 |
| | | | B12. Fixed asset investment per capita (10000 Yuan) | 0.4237 |
| | B2. Industrial structure | 0.2565 | B21. Proportion of secondary industry (%) | 0.2429 |
| | | | B22. Proportion of tertiary industry (%) | 0.7571 |
| | B3. Foreign trade | 0.4892 | B31. Import and export as a share of GDP (%) | 0.6728 |
| | | | B32. Foreign direct investment as a share of GDP (%) | 0.3272 |
| C. Tourism industry | C1. Dependency on tourism income | 0.1755 | C11. Dependency on domestic tourism (%) | 0.2826 |
| | | | C12. Dependency on inbound tourism (%) | 0.7174 |
| | C2. Tourist reception scale | 0.2462 | C21. Domestic travel density (%) | 0.3849 |
| | | | C22. Inbound travel density (%) | 0.6151 |
| | C3. Benefits of tourism industry | 0.2087 | C31. Benefits of travel agencies (10000 Yuan per person) | 0.5877 |
| | | | C32. Benefits of star hotels (10000 Yuan per person) | 0.4123 |
| | C4. Benefits of tourism employment | 0.2199 | C41. Occupational share of travel agency (%) | 0.5516 |
| | | | C42. Occupational share of star hotel (%) | 0.4484 |
| | C5. Tourist behavior | 0.1497 | C51. Consumption of domestic tourism per capita (Yuan/day) | 0.1649 |
| | | | C52. Consumption of inbound tourism per capita (dollar/day) | 0.3536 |
| | | | C53. Average stay of inbound tourism (day) | 0.4815 |

Yearbook, China Tourism Statistical Yearbook, China Environmental Statistical Yearbook and China Regional Economic Statistical Yearbook from 2004 to 2018. The data sources can be found in the Data Availability section and S1 File.

Regarding the indexes of the ecological environment [36, 39–41, 47, 52–56], we attribute the ten basic-class indexes to three first-class indexes: the resource endowment and pressure and response of ecological environment. "Green land, forestland and wetland" are the main carriers of ecological environment construction in China, and so we choose A11, A12 and A13 as the indexes of the ecological environment resource endowment. Pollutant emissions directly affect the quality of the ecological environment, while soil erosion control and pollution control affect the current situation of improving the ecological environment. Therefore, we choose five kinds of pollutant emissions to form an index system for ecological environmental pressure and choose the soil erosion control area and the industrial pollution control investments to reflect the response of an area to an improved ecological environment.

Concerning the indexes of the economic development level [36, 47, 52–54, 57], we attribute the six basic-class indexes to three first-class indexes: the economic level, the industrial structure and foreign trade. The most direct index of the economic level is per capita GDP. The industrial structure is also an important index of regional economic development. The higher the proportions of secondary industry and tertiary industry are, the higher the economic development level. In addition, foreign trade is also an important pillar driving regional economic development.

Concerning the indexes of the tourism industry [36, 44, 47, 55], we attribute the eleven basic-class indexes to five first-class indexes: the dependency on tourism income, the tourist reception scale, the benefits of tourism industry, the benefits of tourism employment and the tourist behavior. For example, the impact of tourists on the local social culture and ecological environment are obvious, but the extent and scope of these impacts are closely related to the density of tourists. The greater the density is, the greater the impact. Therefore, we use the ratio of the number of domestic tourists and the number of inbound tourists to the number of local residents to measure the scale of tourism reception. The industrial effects and employment benefits of travel agencies and star hotels are also important indicators for measuring the tourism industry. It should be noted that the statistics of tourism enterprises were mainly travel agencies, hotels and other statistics before 2009. In 2010, the statistics of tourism enterprises were divided into travel agencies, star hotels and tourist attractions. For the consistency of the time series, only travel agencies and star hotels were selected.

## 2.2 Data standardization

As seen from Table 1, the three subsystems considered in this study all contain multiple evaluation indexes. Since the units and magnitudes of each index are different, and each index has positive or negative effect on the system, these indexes cannot be directly used or compared. Therefore, it is necessary to standardize all indexes to eliminate the influence of units, magnitudes and types on subsystems. The standardized calculation formula that is adopted by this study is as follows:

$$
x'_{ijt} = \begin{cases} \dfrac{x_{ijt} - \min\limits_{i}\{x_{ijt}\}}{\max\limits_{i}\{x_{ijt}\} - \min\limits_{i}\{x_{ijt}\}}, & j \in J^+ \\[4mm] \dfrac{\max\limits_{i}\{x_{ijt}\} - x_{ijt}}{\max\limits_{i}\{x_{ijt}\} - \min\limits_{i}\{x_{ijt}\}}, & j \in J^- \end{cases} \tag{1}
$$

where $x_{ijt}$ represents the sample value of the j-th index of the i-th research province at time t; $J^+$

and $\Gamma$ represent the sets of positive and negative indexes, respectively; $\min_i\{\cdot\}$ and $\max_i\{\cdot\}$ respectively represent the minimum and maximum values of the given j-th index in all research provinces at time t. Obviously, the standardized $x'_{ijt}$ is between 0 and 1. In addition, the research province here refers to 31 provinces in China, and the research period is from 2003 to 2017.

## 2.3 Comprehensive evaluation method

Before calculating the comprehensive evaluation values of the three subsystems, it is necessary to calculate the comprehensive evaluation values of the first-class indexes in each subsystem. Using the standardized data of the basic-class indexes and the appropriate weights, the comprehensive evaluation value of the first-class indexes can be calculated. It is very important to choose or set the weights when calculating the comprehensive values of the first-class indexes and that of the three subsystems. Due to the inevitable subjectivity of the subjective weighting method or expert weighting method, this paper adopts the entropy method as the objective weighting method to calculate the weight, which can avoid the defects of the subjective weighting method to some extent.

The comprehensive evaluation of the first-class indexes includes three steps: data standardization, weight calculation and comprehensive value calculation. The data standardization of the basic-class indexes is shown in section 2.2. The steps to obtain the weights of the basic-class indexes using the entropy method [39–40] are as follows:

$$p_{ijt} = x'_{ijt}/\sum_i x'_{ijt}, \quad j \in S_k \tag{2}$$

$$e_{jt} = -\delta \sum_i p_{ijt} \ln p_{ijt}, \quad j \in S_k \tag{3}$$

$$w_{jt} = (1 - e_{jt})/\sum_j (1 - e_{jt}), \quad j \in S_k \tag{4}$$

where $S_k$ represents a set of basic-class indexes belonging to the given first-class index k, and $p_{ijt}$ indicates the proportion of the basic-class index j that is subject to the first-class index k of the i-th province at time t. For example, for the ecological environment endowment, we should only consider three basic-class indexes: wetland area per capita, the forest cover rate and green areas per capita. $e_{jt}$ represents the information entropy of index j at time t. The constant $\delta$ is calculated by $\delta = 1/\ln m$ and m is the number of samples. Then we have $0 \leq e_{jt} \leq 1$. $w_{jt}$ indicates the weight of index j at time t. The weight results of each basic-class index of the three subsystems from 2003 to 2017 can be found in the Data Availability section and S2 File. The column 5 of Table 1 give the average entropy weights of each basic-class index of the three subsystems.

After calculating the entropy weights of the basic-class indexes, the comprehensive evaluation values of each first-class index of the three subsystems are calculated as follows:

$$y_{ikt} = \sum_j w_{jt} x'_{ijt}, \quad j \in S_k \tag{5}$$

The comprehensive evaluation value of the three subsystems also includes three steps: data standardization, weight calculation, and comprehensive value calculation. Among them, the calculation steps and formulas of the data standardization and entropy weights are identical to those of the first-class indexes. The weight results of each first-class index of the three

subsystems from 2003 to 2017 can be found in the Data Availability section and S2 File. The column 3 of Table 1 give the average entropy weight of each first-class index. The comprehensive evaluation formulas of the three subsystems are as follows.

$$U_{\text{envi},it} = \sum_k \lambda_{ikt} y'_{ikt}, \quad k \in A = \{A1, A2, A3\} \tag{6}$$

$$U_{\text{econ},it} = \sum_k \lambda_{ikt} y'_{ikt}, \quad k \in B = \{B1, B2, B3\} \tag{7}$$

$$U_{tour,it} = \sum_k \lambda_{ikt} y'_{ikt}, \quad k \in C = \{C1, C2, C3, C4, C5\} \tag{8}$$

where, $A = \{A1, A2, A3\}$, $B = \{B1, B2, B3\}$ and $C = \{C1, C2, C3, C4, C5\}$ represent the sets of first-class indexes for the three subsystems, respectively. $U_{\text{envi},it}$, $U_{\text{econ},it}$ and $U_{tour,it}$ represent the comprehensive evaluation values of the ecological environment, the economic development level and the tourism industry subsystem of the i-th province at time t, respectively. $y'_{ikt}$ and $\lambda_{ikt}$ are the standardized evaluation value and the entropy weight of the first-class index k of the i-th province at time t, respectively.

## 2.4 Coupling coordination model

After calculating the comprehensive evaluation values of the three subsystems, we can calculate the comprehensive evaluation values of the whole system of 31 provinces in each period. The comprehensive evaluation value is also the comprehensive coordination index of the coupling coordination model, and the formula is as follows:

$$T_{it} = w_{\text{envi},it} \cdot U'_{\text{envi},it} + w_{\text{econ},it} \cdot U'_{\text{econ},it} + w_{tour,it} \cdot U'_{tour,it} \tag{9}$$

where, $U'_{\text{envi},it}$, $U'_{\text{econ},it}$ and $U'_{tour,it}$ are the standardized results of $U_{\text{envi},it}$, $U_{\text{econ},it}$ and $U_{tour,it}$, respectively, and they satisfy $w_{\text{envi},it} + w_{\text{econ},it} + w_{tour,it} = 1$ and $w_{\text{envi},it} + w_{\text{econ},it} + w_{tour,it} \geq 0$.

The above formula for calculating the comprehensive coordination index includes the determination of the weights $w_{\text{envi},it}$, $w_{\text{econ},it}$ and $w_{tour,it}$. Some researchers believe that the importance of each subsystem is the same and thus set average weight; otherwise subjective weights are set (for example, the three weights can be set as 0.4, 0.4 and 0.2 respectively). We believe that average weights ignore the importance of different subsystems, and subjective weights make it difficult to compare the final results due to the subjectivity of the researchers or users. Therefore, the entropy method is adopted to calculate the weights of the three subsystems in each period. The weight results of the three subsystems in China from 2003 to 2017 can be found in the Data Availability section and S2 File. The average entropy weight of the tourism industry is approximately 0.2. The entropy weight of the economic development level fluctuates approximately 0.24, while that of the ecological environment fluctuates approximately 0.55. This shows that the importance of the three subsystems is not the same when considering the coupling and coordinated relationships of the whole system, especially when the role of the ecological environment in the whole system exceeds those of the economic development level and tourism industry.

The concept of the coupling degree comes from capacity coupling in physics, which refers to the dynamic relationship between subsystems that are interdependent and interact [40]. It reveals the phenomenon that multiple subsystems influence each other and even cooperate through various interactions. The mathematical formula of n subsystems, which describes the

coupling degree, is as follows:

$$C_n = \left[\frac{U_1 U_2 \cdots U_n}{\prod_{i \neq j}(U_i + U_j)}\right]^{1/n} \tag{10}$$

where, $U_i$ represents the comprehensive value of the i-th subsystem, and $n$ represent the total number of subsystems. The larger that the value of the coupling degree $C_n$ is, the better the coupling effect between subsystems is, and the more obvious the correlation effect is. It is not difficult to prove that the range of the coupling degree $C_n(n > 2)$ that is mentioned above is within [0, 0.5] [54]. Therefore, for the case of the three subsystems considered in this paper, the specific form of the coupling degree calculation formula is as follows:

$$C_{it} = 2 \cdot \sqrt[3]{\frac{U_{envi,it} \times U_{econ,it} \times U_{tour,it}}{(U_{envi,it} + U_{econ,it})(U_{envi,it} + U_{tour,it})(U_{econ,it} + U_{tour,it})}} \tag{11}$$

where $C_{it}$ represents the coupling degree of the i-th province at time t.

The coupling degree can reflect the strength of the interaction among subsystems, but it cannot reflect the overall coordination of large-scale systems. Therefore, the coordination index reflecting the comprehensive development level is introduced, and its calculation formula is as follows:

$$D_{it} = \sqrt{C_{it} \times T_{it}} \tag{12}$$

where, $D_{it}$ represent the coordination degree of the i-th province at time t.

After calculating the coupling and coordination degrees, the coupling level and coordination level can be divided into multiple categories according to their numerical value, and they are usually divided into four categories: low level, medium level, high level and extremely high level. Many scholars use subjective threshold values to classify the coupling level or coordination level. This method has two defects. One is that the decision-making basis of each decision-maker is different and the results are not comparable. In addition, when the calculated coupling or coordination degree shows an uneven distribution, the subjective division tends to cause classification anomalies or inaccuracies. This study does not quantitatively and subjectively divide the categories. Instead, it chooses the objective quartile method so that the coupling and coordination classifications are more objective. It also helps to divide the 31 provinces in China into four categories according to their coupling degrees and coordination degrees. In addition, this also satisfies the four categories of the coupling degree and coordination degree that were mentioned before.

## 3 Empirical results

### 3.1 Temporal characteristics of coupling and coordination degrees

After establishing the evaluation index system of the ecological environment-economic development level-tourism industry, the weights are calculated based on the entropy method. First, the comprehensive value of the first-class indexes is calculated by using the sample data of the basic-class indexes, and then the comprehensive values of the three subsystems are calculated by using the comprehensive value of the first-class indexes. Finally, the coupling degree and coordinating degree between the three subsystems are calculated according to the coupling degree formula and the coordination degree formula. Accordingly, the average values of the coupling and coordination degree of 31 provinces in China from 2003 to 2017 are analyzed.

The final calculation results of coupling and coordination model from 2003 to 2017 are shown in S3 File.

As seen from Fig 1, the average coupling degree and coordination degree of the 31 provinces in China are between 0.858–0.883 and 0.286–0.324, respectively, both of which fluctuate upward and downward. In other words, as time passes, the interaction between the three subsystems increases, and the coordinated relationship improves. Other scholars have also conducted some related research. Zhou et al. analyzed the coordinated development of the Yangtze Economic Zone of China and the results showed that the coordination degree of the tourism-economy-ecological environment system experienced stable or fluctuant increases [52]. Jiang et al. analyzed the evolution of coordination degrees of the economy-resource-environment system and the results showed that the coupling degree and coordination degree of China have increased year by year in 2003–2014 [54]. Chen et al. analyzed the temporal and spatial evolution of the coupling coordination development of the tourism-ecological environment system and the results showed that the coupling degree and coordination degree steadily increased from 2007–2016 [55]. Those situations are basically consistent with the conclusions of this study.

With respect to the coupling degree, the coupling degree shows an upward trend over every five-year period of China. Because of the heavy snow in the south China in 2008 and the global financial crisis, the coupling degree in 2009 hit its obviously lowest point, which makes its interaction force poor.

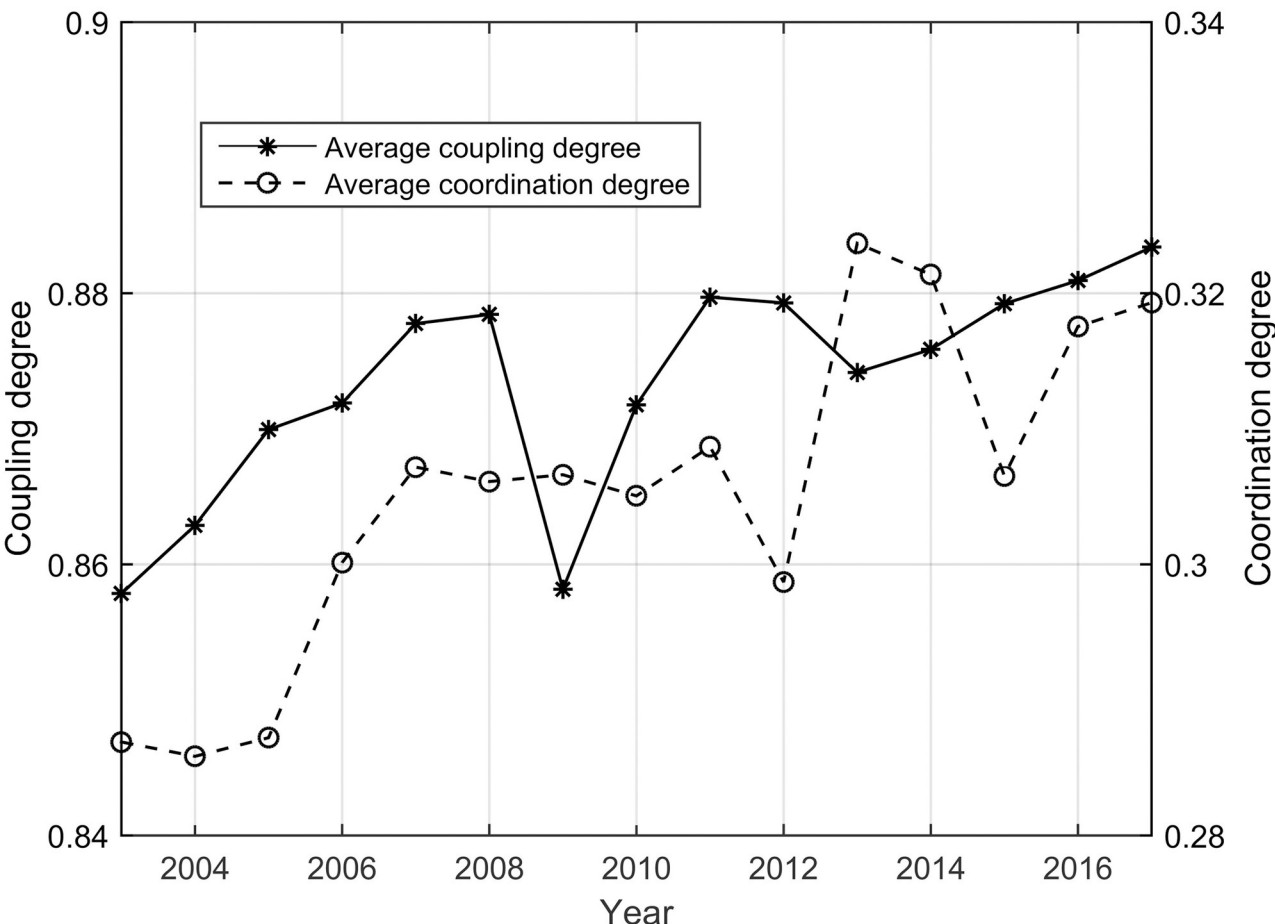

**Fig 1. Average values of the coupling and coordinating degree during 2003–2017.**

With respect to coordination, during the tenth five-year plan period of China (2003–2005), the coordination effect of the three subsystems is not obvious, and the coordination value is the lowest. During the 11th and 12th five-year plan period (2006–2010 and 2011–2015, respectively), the coordination of the three subsystems continued to improve, reaching the highest coordination in 2013. Only in 2012 and 2015 did the coordination between the three subsystems hit a reduced turning point. During the 13th five-year plan period (2016–2017), the coordination of the three subsystems slows down, but it still shows an increasing trend.

### 3.2 Spatial distribution of coupling and coordination degrees

We calculate the coupling degree during the period from 2003–2017 and classify the coupling degree of each year according to the quartile method. The 31 provinces are classified into four types: namely, low level coupling, medium level coupling, high level coupling and extremely high level coupling. The first five figures in Fig 2 show the spatial distributions of the coupling degrees of the 31 provinces in the two years before and after the 10th, 11th, 12th and 13th five-year plan periods in China.

It can be seen from Fig 2 that only Fujian Province changed from medium level coupling to extremely high coupling during the 10th five-year plan period (2003–2005), but Fujian Province changed from extremely high coupling to medium coupling during the 11th five-year plan period (2006–2010). During the 12th and 13th five-year plan periods (2011–2015 and 2016–2017, respectively), there are significant changes in the coupling levels of the two provinces. Shanxi and Henan fell from the extremely high level to medium level during the 12th five-year plan period. Shanxi and Shandong rose from medium level to extremely high level coupling during the 13th five-year plan period.

From the spatial distribution of the coupling degree in 2003, the four central and four western provinces, namely, Jilin, Jiangxi, Henan, Hunan, Guangxi, Sichuan, Shaanxi and Xinjiang have extremely high coupling degrees. In 2003, several provinces belong to the low coupling level of the comprehensive index of the three subsystems. Especially in Henan Province, the comprehensive evaluation index of the three subsystems was ranked the second from the bottom, while Jilin, Xinjiang and Jiangxi were ranked at the top. Most of the provinces with low coupling degrees are concentrated in Beijing, Shanghai, Jiangsu, and Guangdong, which have good economic development and good tourism development but poor ecological environment. The provinces with low coupling degree also include Tibet, Guizhou, Yunnan and Qinghai, which have poor economic development or late tourism development.

From the spatial distribution of the coupling degree in 2017, extremely high coupling values are still concentrated in the four central provinces and two eastern provinces with poor economic development and poor ecological environment. The provinces with extremely high coupling also include two western provinces that have poor economies and slow tourism but better ecological environments. These provinces include Jilin, Jiangxi, Hubei, Shanxi, Shandong, Hebei, Ningxia and Xinjiang. The provinces with low coupling degrees are mainly distributed in the three eastern provinces with good economic development and rapid tourism development but poor ecological environments, and three western provinces with median economic and tourism development, specifically Beijing, Tianjin, Shanghai, Guangdong, Chongqing, Tibet and Qinghai.

To understand the coordinated relationship between subsystems, further analysis of the coordination degree is needed. Therefore, we calculate the coordination degrees of the 31 provinces from 2003 to 2017 and classify the coordination degree of each year according to the quartile method. The 31 provinces are classified into four types: low level coordination, medium level coordination, high level coordination and extremely high level coordination.

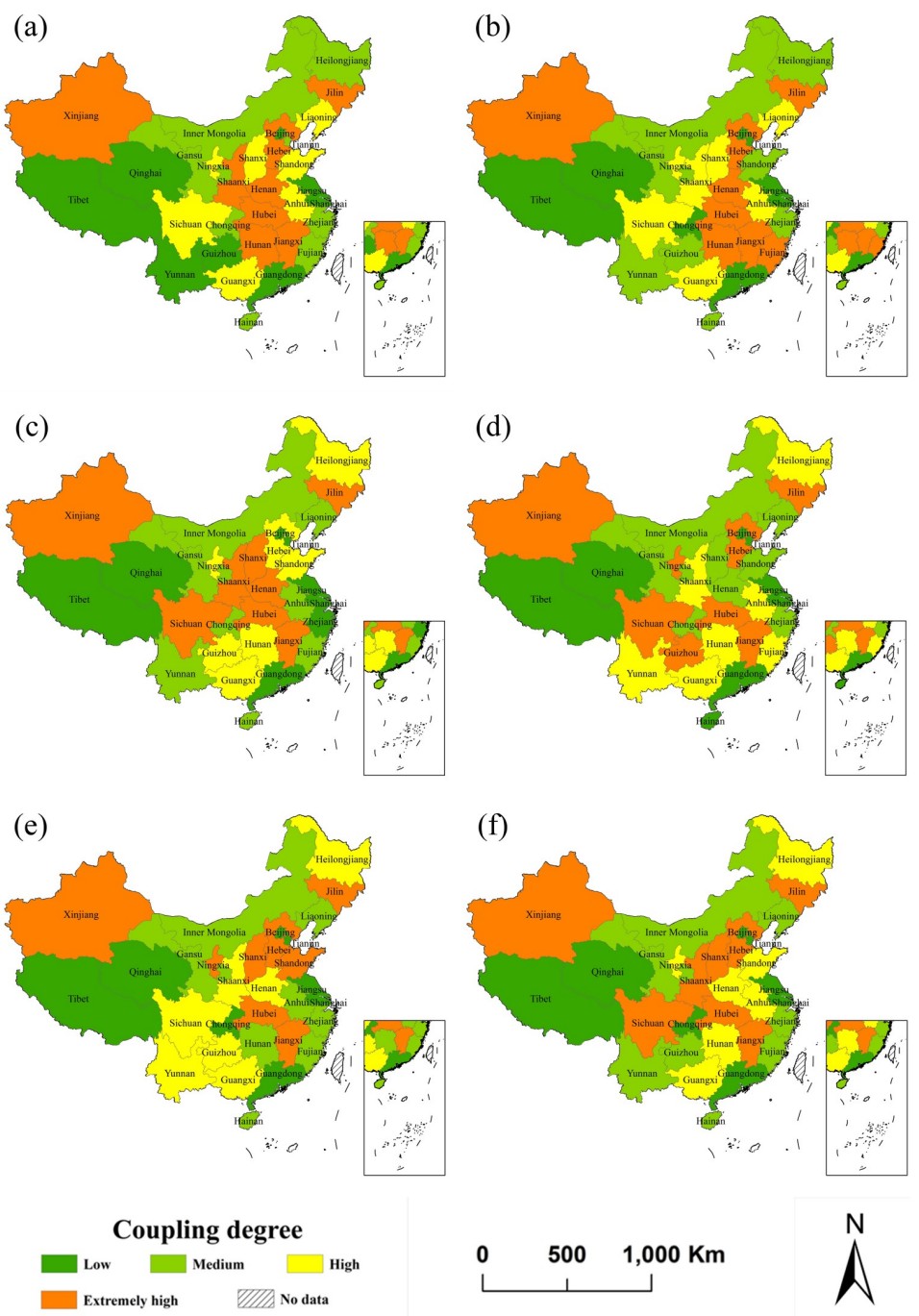

**Fig 2. Coupling degree distribution of the 31 provinces in China at time 2003(a), 2005(b), 2010(c), 2015(d), 2017 (e) and the average from 2003-2017(f).**

The first five figures in Fig 3 show the spatial distributions of the coordination degree of 31 provinces in the two years before and after the 10th, 11th, 12th and 13th five-year plan period in China. It can be seen from Fig 3 that only Xinjiang's coordination level has significantly changed from high level to low level coordination and from low level to high level coordination in the 11th and 12th five-year plan periods respectively. In addition, during the four

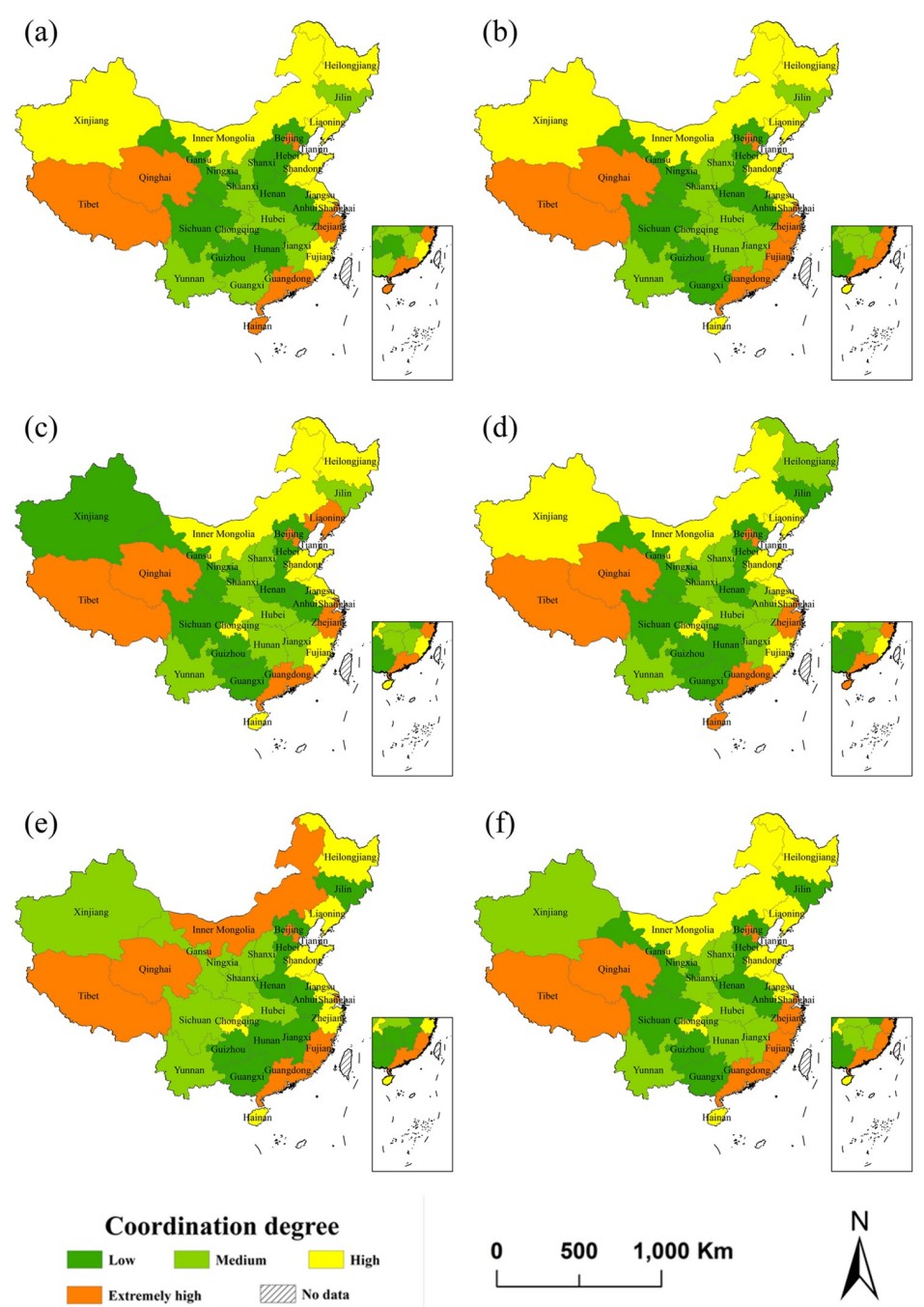

**Fig 3. Coordination degree distribution of 31 provinces in China at time 2003(a), 2005(b), 2010(c), 2015(d), 2017 (e) and the average from 2003-2017(f).**

five-year plan periods, the coordination level of 13 provinces increased slightly, while the coordination level of 13 provinces decreased slightly.

From the coordination levels in 2003 and 2017, the extremely high level coordinated provinces are mainly in Beijing, Tianjin and Guangdong, which have rapid economic and tourism development but poor ecological environment, and Tibet and Qinghai, which have poor economic and tourism development but very good ecological environment. In addition, Zhejiang

and Hainan in 2003 and Shanghai and Fujian in 2017 all have extremely high level coordination. Hebei, Anhui, Henan, Hunan and Guizhou, all of which have poor economic, tourism and ecological environment development, have low level coordination both in 2003 and in 2017. Shanxi, Sichuan, and Gansu have low coordination levels in 2003, while Jilin, Jiangxi, and Guangxi have low coordination level in 2017.

The distribution of the coordination degree in each province is subject to the comprehensive evaluation indexes and weights of the three subsystems. For example, the economic development levels and tourism industries of Beijing and Tianjin rank among the top five in the country, while their ecological environments rank relatively low. These two provinces have extremely high levels of coordination both in 2003 and 2017. Qinghai is a typical province with backward economic and tourism development but an excellent ecological environment. As the entropy weight of the ecological environment subsystem is approximately 0.55, Qinghai is also a province with extremely high coordination levels in 2003 and 2017.

From the last figures of Figs 2 and 3, we can see that the average coupling and coordination degrees of Guizhou, Gansu and Yunnan are relatively low, while the average coupling and coordination degrees of Heilongjiang and Shandong are relatively high. The coastal provinces, such as Beijing, Tianjin, Shanghai, Jiangsu, Zhejiang, Guangdong, Fujian and Chongqing, with developed economies and high ecological environment pressure have relatively low level coupling but extremely high level coordination. Many provinces of central and western China with good ecological environment protection and economic backwardness have high level coupling and low level coordination. This is because a developed economy can promote the coordination level in economically developed provinces to a certain extent, but the scarcity of natural resource endowments, large number of inhabitants and large pollutant emissions cause the ecological environment to deteriorate, resulting in low level coupling. The underdeveloped economies decrease the coordination levels of central and western China, but the high natural resource endowments and the lower pollutant emissions lead to high scores for the ecological environment, which ultimately increases the coupling degree in these provinces. Other scholars also conducted some relevant studies. A representative example is Zhou's researches. The coordination degree reached a high level to the east and a low level to the west of the Yangtze River Economic Zone [52], and the coordination degrees in the coastal provinces exceed those in the western and central provinces of China [53]. This corresponds with the results of this study.

### 3.3 Analysis of development type

Since the ecological environment has a great impact on economic development and tourism development, and the weight of the ecological environment exceeds the weights of the economy and tourism, most provinces in China have high level coupling but low level coordination or low level coupling but high level coordination. To further understand the reasons for the strong interaction but poor coordination or weak interaction but strong coordination among the three subsystems, the 31 provinces in China are further classified and analyzed.

From the perspective of ecological environment development, the economic development level and the tourism industry, the average comprehensive evaluation values of the three subsystems of the 31 provinces in China are compared, and the development of the 31 provinces can be classified into four basic types: (1) an advanced economy type with basic coordination between the economy, tourism and ecological environment; (2) an advanced tourism type with basic coordination between the economy, tourism and the ecological environment; (3) an advanced economy type with economic and tourism development beyond the bearing capacity of the ecological environment; and (4) an advanced tourism type with economic and tourism

development beyond the bearing capacity of the ecological environment. In Fig 4, these four types are represented by advanced economy and basic coordination, advanced tourism and basic coordination, advanced economy but exceed ecological capacity and advanced tourism but exceed ecological capacity, respectively.

It can be seen from Fig 4 that the provinces where the development of the ecological environment, economy and tourism system are basically coordinated are distributed in seven western provinces and three central provinces. Among them, Inner Mongolia, Gansu, Qinghai, Ningxia, Jilin and Jiangxi are in extensive economic development modes with imperfect tourism facilities and low tourism dependence. It is necessary to make full use of the good ecological environment and develop characteristic tourism products to accelerate tourism development. Heilongjiang, Tibet, Xinjiang and Shanxi should make full use of their frontier advantages and unique tourism resources, integrate their primary, secondary and tertiary industries and local culture, and give full play to the driving role of tourism.

The remaining 21 provinces in China are in states of development that sacrifice or overutilize the ecological environment. Among them, eight of the eleven eastern provinces rely on

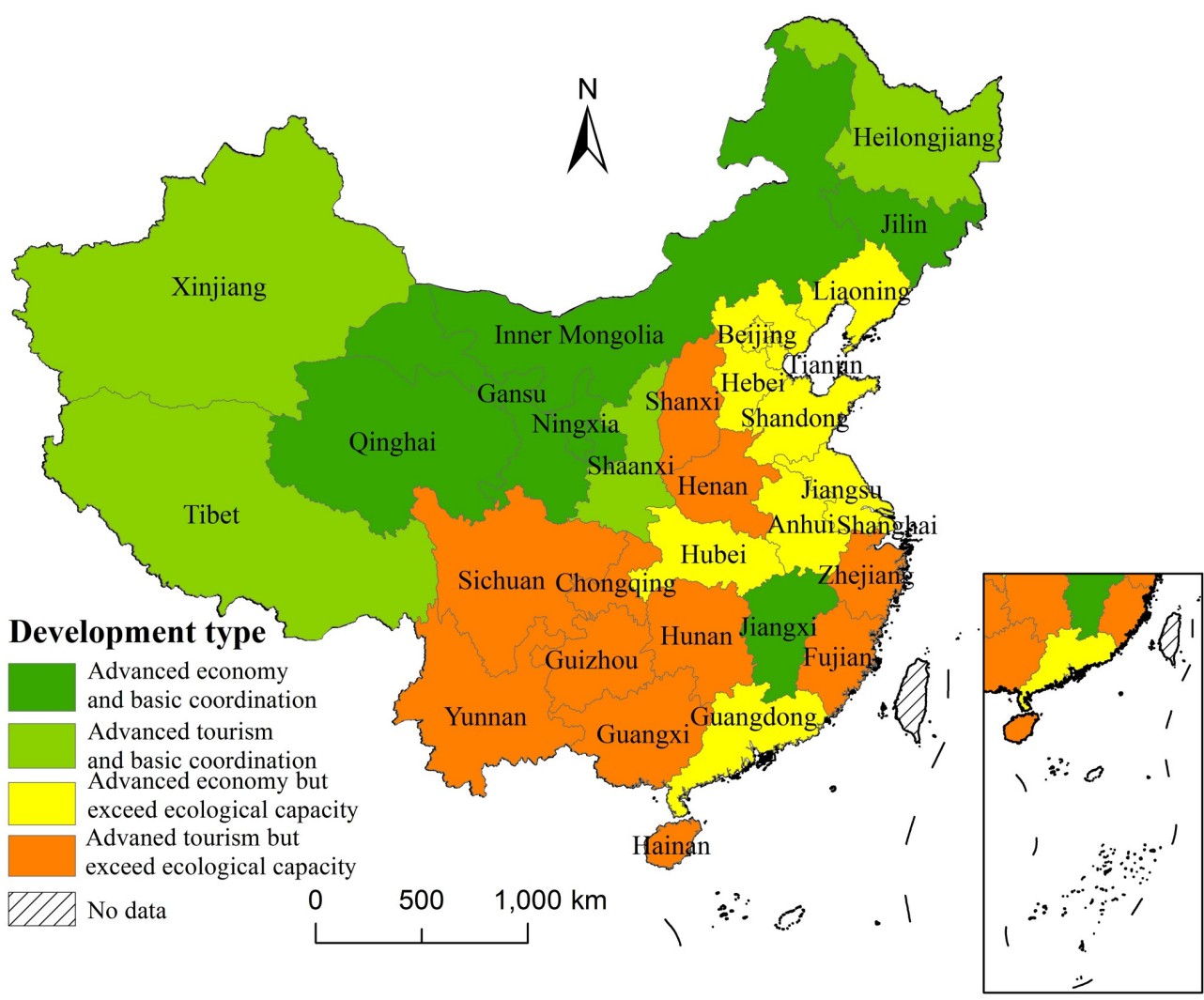

**Fig 4. Development type of the 31 provinces in China.**

the excessive consumption of ecological resources for economic development, and four of the five southwestern provinces over-use tourism resources, resulting in serious overloads of their ecological environments. The results of Zhou et al. show that the eastern coastal provinces have economic and tourism advantages, but their ecological environments are poorly protected. Although the economically weak western provinces have good ecological environment protection, their tourism is relatively backwards [53]. It is basically consistent with the results of this study.

## 4 Conclusions and policy recommendations

This study establishes a comprehensive evaluation index system of the ecological environments, the economic development levels and the tourism industries of 31 provinces in China. The entropy method is used to calculate the weights, and the coupling coordination model is used to analyze the coupling and coordinated relationships of the three subsystems of 31 provinces.

From the perspective of the provincial coupling degrees at the beginning and the end of the study periods, the changes in the coupling degrees in most provinces is not obvious. The high coupling provinces are mainly in the central and western provinces where the economic and tourism development are backward or the ecological environment is poor. The low coupling provinces are mainly distributed in the eastern and western provinces where the economic development is rapid and the tourism industry is good while the ecological environment is poor.

From the perspective of the provincial coordination degrees at the beginning and the end of the study period, Beijing, Tianjin, and Guangdong, which have good economic development and rapid tourism development but poor ecological environments, have extremely high level coordination. Tibet and Qinghai, which have backward or poor economic and tourism development but very good ecological environments, also have extremely high level coordination. Hebei, Anhui, Henan, Hunan and Guizhou provinces with poor economic, tourism and ecological environment development are all low level coordination provinces.

From the perspective of the average coupling and coordination degrees, the average coupling and coordination degrees of Guizhou, Gansu and Yunnan are relatively low, while those of Heilongjiang and Shandong are extremely high. Chongqing and the coastal provinces with high ecological environment pressure but good economic development have low coupling degree and extremely high coordination degrees. The vast central and western provinces with good ecological environment protection and economic backwardness have high coupling degrees and low coordination degrees.

According to the comprehensive evaluation index of the subsystems, the economies and tourism industries of all eleven eastern provinces exceeded the carrying threshold of the ecological environment. Five eastern provinces and five southwestern provinces are over-reliant on the ecological environment for tourism development, resulting in the economic and tourism development exceeding the threshold of the ecological environment. The economic and tourism development in most western provinces and a few central provinces are basically coordinated with the ecological environments.

At the national level, we need to focus on promoting the development of high-quality tourism in the central and western provinces of China. For example, in Gansu, Qinghai, Ningxia, Heilongjiang and Xinjiang, where the economy, tourism and ecological environment are basically in harmony with the backward development of tourism. We should make full use of the high-quality tourism resources and good ecological environment to further improve the tourism infrastructure and the tourism reception capacity. Xinjiang should build the core area of

the Silk Road, Ningxia should build a good inland open economic pilot area, Heilongjiang should develop border tourism and international tourism by speeding up the improvement of railway corridors and regional railway network to Russia. Five southwestern provinces should make good use of high-endowment tourism resources and develop excellent tourism resource products while protecting the ecological environment. Chongqing should develop an important support model for the western provinces of China. Guangxi should make good use of the opportunity of the Belt and Road Initiative and develop the economic and tourism cooperation with the southwest and central and southern provinces. Yunnan should make good use of the new plateau of Greater Mekong Subregional Economic Cooperation and built the radiation center of South Asia and Southeast Asia. Eleven eastern provinces are developing rapidly in economy and tourism. We should pay more attention to protecting the ecological environment and developing high-quality tourism products.

## Supporting information

**S1 File. Origin data of three subsystems of 31 provinces in China from 2003 to 2017.** (RAR)

**S2 File. All weight results calculated by entropy method.** (XLSX)

**S3 File. The final calculation results of coupling and coordination model.** (XLSX)

**S1 Fig.** (XLSX)

**S2 Fig.** (XLSX)

## Author Contributions

**Conceptualization:** Zheng Wang.

**Data curation:** Haibin Xia, Yanlin Yue.

**Methodology:** Zhizhu Lai, Dongmei Ge.

**Writing – original draft:** Zhizhu Lai, Dongmei Ge.

**Writing – review & editing:** Zhizhu Lai, Dongmei Ge, Haibin Xia, Yanlin Yue, Zheng Wang.

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
