## [Editor Report · Decision Letter 0]

14 Oct 2019

PONE-D-19-27598

Coupling coordination analysis of environment, economy and tourism：A case study of China

PLOS ONE

Dear Pro. wang,

Thank you for submitting your manuscript to PLOS ONE. After careful consideration, we feel that it has merit but does not fully meet PLOS ONE’s publication criteria as it currently stands. Therefore, we invite you to submit a revised version of the manuscript that addresses the points raised during the review process.

We would appreciate receiving your revised manuscript by Nov 28 2019 11:59PM. To enhance the reproducibility of your results, we recommend that if applicable you deposit your laboratory protocols in protocols.io, where a protocol can be assigned its own identifier (DOI) such that it can be cited independently in the future. For instructions see: http://journals.plos.org/plosone/s/submission-guidelines#loc-laboratory-protocols

We look forward to receiving your revised manuscript.

Kind regards,

Bing Xue, Ph.D.

Academic Editor

PLOS ONE

**Journal Requirements:**

2. Our editorial staff has assessed your submission, and we have concerns about the grammar, usage, and overall readability of the manuscript.  We therefore request that you revise the text to fix the grammatical errors and improve the overall readability of the text before we send it for review. We suggest you have a fluent, preferably native, English-language speaker thoroughly copyedit your manuscript for language usage, spelling, and grammar.

If you do not know anyone who can do this, you may wish to consider employing a professional scientific editing service.  

Whilst you may use any professional scientific editing service of your choice, PLOS has partnered with both American Journal Experts (AJE) and Editage to provide discounted services to PLOS authors. Both organizations have experience helping authors meet PLOS guidelines and can provide language editing, translation, manuscript formatting, and figure formatting to ensure your manuscript meets our submission guidelines. To take advantage of our partnership with AJE, visit the AJE website (http://learn.aje.com/plos/) and enter referral code PLOS15 for a 15% discount off AJE services. To take advantage of our partnership with Editage, visit the Editage website (www.editage.com) and enter referral code PLOSEDIT for a 15% discount off Editage services. If the PLOS editorial team finds any language issues in text that either AJE or Editage has edited, the service provider will re-edit the text for free.

Please note that PLOS ONE does not copyedit accepted manuscripts and that one of our criteria for publication is that articles must be presented in an intelligible fashion and written in clear, correct, and unambiguous English (http://www.plosone.org/static/publication#language). If the language is not sufficiently improved, we may have no choice but to reject the manuscript without review.

3. We note that  Figure(s) 3,4,& 5 in your submission contain [map/satellite] images which may be copyrighted. All PLOS content is published under the Creative Commons Attribution License (CC BY 4.0), which means that the manuscript, images, and Supporting Information files will be freely available online, and any third party is permitted to access, download, copy, distribute, and use these materials in any way, even commercially, with proper attribution. For these reasons, we cannot publish previously copyrighted maps or satellite images created using proprietary data, such as Google software (Google Maps, Street View, and Earth). For more information, see our copyright guidelines: http://journals.plos.org/plosone/s/licenses-and-copyright.

a) You may seek permission from the original copyright holder of Figure(s) [#] to publish the content specifically under the CC BY 4.0 license.  

**Additional Editor Comments (if provided):**

This is an interesting paper, however, some major revisions are required before it was considered for going to external review.

1. I think that the authors have not adequately described their analysis in this original manuscript, for example, how the authors obtained datasets, any parameters used, and please keep in mind that, if any existing datasets were used for testing, it should have enough details for another researcher to reproduce the findings. A data-source should be presented in the context or as a supp. file.

2. References are not sufficient. At least, please add some relevant references in the first Paragraph of the introduction section. And a literature review is suggested to be added.

3. The language should be improved.

4. Policy implications should be considered as a solid section before the conclusions.

5. Please try your best to concentrate your findings from the modelling analysis, and add some discussions by such as comparative analysis with the external outcomes from other scholars.

6. I'd like to see the revision.

---

## [Author Response · Author response to Decision Letter 0]

16 Nov 2019

Q1. I think that the authors have not adequately described their analysis in this original manuscript, for example, how the authors obtained datasets, any parameters used, and please keep in mind that, if any existing datasets were used for testing, it should have enough details for another researcher to reproduce the findings. A data-source should be presented in the context or as a supp. file. 

Answer: We made a detailed description of the selection of data sources and indicators in the revised version, and added the datasets as an attachment to the revised version.

Q2. References are not sufficient. At least, please add some relevant references in the first Paragraph of the introduction section. And a literature review is suggested to be added.

Answer: We have added references and literature reviews to the introduction.

Q3. The language should be improved. 

Answer: We have employed the AJE (http://learn.aje.com) to thoroughly copyedit this manuscript for language usage, spelling, and grammar.

Q4. Policy implications should be considered as a solid section before the conclusions. 

Answer: We rewrote the policy recommendations and placed them separately in Section 3.4.

Q5. Please try your best to concentrate your findings from the modelling analysis, and add some discussions by such as comparative analysis with the external outcomes from other scholars.

Answer: We have added some comparative discussions with other literature results in the empirical analysis section.

---

## [Decision Letter · Decision Letter 1]

18 Dec 2019

PONE-D-19-27598R1

Coupling coordination between environment, economy and tourism：A case study of China

PLOS ONE

Dear Pro. wang,

Thank you for submitting your manuscript to PLOS ONE. After careful consideration, we feel that it has merit but does not fully meet PLOS ONE’s publication criteria as it currently stands. Therefore, we invite you to submit a revised version of the manuscript that addresses the points raised during the review process.

We would appreciate receiving your revised manuscript by Feb 01 2020 11:59PM. To enhance the reproducibility of your results, we recommend that if applicable you deposit your laboratory protocols in protocols.io, where a protocol can be assigned its own identifier (DOI) such that it can be cited independently in the future. For instructions see: http://journals.plos.org/plosone/s/submission-guidelines#loc-laboratory-protocols

We look forward to receiving your revised manuscript.

Kind regards,

Bing Xue, Ph.D.

Academic Editor

PLOS ONE

Reviewers' comments:

Reviewer's Responses to Questions

**Comments to the Author**

1. If the authors have adequately addressed your comments raised in a previous round of review and you feel that this manuscript is now acceptable for publication, you may indicate that here to bypass the “Comments to the Author” section, enter your conflict of interest statement in the “Confidential to Editor” section, and submit your "Accept" recommendation.

Reviewer #1: All comments have been addressed

Reviewer #2: (No Response)

Reviewer #3: (No Response)

2. Is the manuscript technically sound, and do the data support the conclusions?

Reviewer #1: Yes

Reviewer #2: Yes

Reviewer #3: Yes

3. Has the statistical analysis been performed appropriately and rigorously? 

Reviewer #1: Yes

Reviewer #2: N/A

Reviewer #3: Yes

4. Have the authors made all data underlying the findings in their manuscript fully available?

Reviewer #1: Yes

Reviewer #2: Yes

Reviewer #3: Yes

5. Is the manuscript presented in an intelligible fashion and written in standard English?

Reviewer #1: Yes

Reviewer #2: No

Reviewer #3: Yes

6. Review Comments to the Author

Reviewer #1: This paper used a comprehensive evaluation index system to evaluate ecological environment, economy and tourism subsystems and then applied coupling degrees and coordination degrees to calculate the relationships between them. Overall, the topic of this research is interesting and draws readers great attention. However, it still suffers little drawbacks. I would like to offer my concerns, which I hope can help to improve the quality of the paper. They are as follows.

1. The introduction part is not sufficient. Especially, the contribution is unclear enough. Hence, the authors should rewrite it.

2. Table 1 needs to be improved. For Table 2,3, and 4, the authors had better convert them to figures in a bid to improve the readability.

3. Policy recommendations should be moved to the end. Besides, it should be rewritten based on the conclusions.

Reviewer #2: This manuscript adopted a coupling coordination model to analyze the relationship between ecological environment, economy and tourism in China’s provinces during 2003-2017. However, it could not be published in its current form.

First, the research contributions are not clear. A potential innovation in this manuscript is its consideration of using coupling model to analysis the relationship between environment, economy and tourism within China. previous papers have included this development and its application in understanding China’s coordination development between economy and tourism and environment. If there is no innovation in the method, what is your most important contribution?

Second, you can refer to some more professional research to standardize the expression, for instance, coupling coordination degree or coordination degree? Tourism is an integral part of the economy, thus, Is it feasible to use “coupling” to describe the relationship between tourism industry and economy.

Third, the manuscript contains statements that are too hard to understand. It's almost like the author(s) had run out of steam at this stage and as the reader, I must say that I felt relieved to have completed the marathon! Specific comments with regard to grammar, flow and basic expression are too numerous to mention. Assistance with writing skills and grammar is needed through an independent third party - someone who knows little about the technical content but can assist with these shortcomings. The paper falls short of being a professionally written management article.

Four, It needs a major rewrite and rethink to eliminate redundant material and unrelated debate and discussion. For example, the sentence in lines 28-30, page 1; the last sentence in 3.4, page 15, etc.

The results are not clearly presented, for example, especial spatial distribution pattern of coupling coordination development for the 31 provinces.

Some technical erros: “they have good representativeness, understanding, availability, integrity and dissemination”(lines 3-4, page 4), pay attention to “understanding”

“are negative indexes (the larger the better), the other indexes are all positive indicators (the smaller the better)”((lines 3-4, page 4), for positive indicators, It should be “the larger the better”

Section 3.4 contains statements that are just far too basic and obvious. Policy recommendations are not targeted, they should be given on the basis of your results specifically.

Reviewer #3: The paper under review studies coordinated development of the ecological environment, the economy and tourism of China from 2003 to 2017. It constructs the comprehensive evaluation index system and uses the coupling coordination model to study the coupling and development of the three subsystems.

1. Line 7 of the first Paragraph in Section 1: “The impact and pressure of this extensive economic development has (should be have) led to…”. There exists an inappropriate tense.

2. Line 1 of the second Paragraph in Section 1: “…, much literature has studied the interaction …”. There exists an inappropriate tense.

3. Lines 1, 2 & 3 of the last Paragraph in Section 1: “…, whether the ecological environment can be coordinated with economic and tourism development has become a problem that many scholars and government departments need to study.”. The narrative here should be improved.

4. The first Section of the paper does not significantly explain what contributions it has made. It is suggested that the main contributions of the paper be listed in the last Paragraph of Section 1.

5. The entropy weighting results of the three subsystems in Section 2.4 are not the focus of the paper. It is recommended to replace Table 1 with a concise language narrative.

6. The results in Sections 3.2 and 3.3 are relatively thorough, but the results expressed in tabular form are worse than the graphical representation. It is recommended to replace Tables 2, 3 and 4 with the corresponding graphical representation.

7. The conclusions of Section 4 are too much and too long. It is recommended to reduce the length and express the main conclusions of the paper in a clear and concise language.

7. PLOS authors have the option to publish the peer review history of their article (what does this mean?). If published, this will include your full peer review and any attached files.

Reviewer #1: No

Reviewer #2: No

Reviewer #3: No

---

## [Author Response · Author response to Decision Letter 1]

8 Jan 2020

Dear Reviewers:

Thank you for your comments concerning our manuscript entitled “ Coupling coordination between environment, economy and tourism：A case study of China” (No.: PONE-D-19-27598R1). Those comments are all valuable and very helpful for revising and improving our paper. We have studied comments carefully and have made correction which we hope meet with approval. The main corrections in the paper and the responds to the reviewer’s comments are as follows.

Responds to the reviewer’s comments:

Reviewer #1: 

1.The introduction part is not sufficient. Especially, the contribution is unclear enough. Hence, the authors should rewrite it. 

Response: Thanks for your suggestions. We have rewritten the introduction in the ‘Manuscript’ (Lines 27-85).

2.Table 1 needs to be improved. For Table 2,3, and 4, the authors had better convert them to figures in a bid to improve the readability. 

Response: Thanks for your suggestions. We have improved the Table 1 in the ‘Manuscript’. The Table 2, 3, and 4 have been converted to the Figure 2, 3, and 4 in the ‘Manuscript’ (Lines 286-288, 345-347, 401-402).

3. Policy recommendations should be moved to the end. Besides, it should be rewritten based on the conclusions. 

Response: Thanks for your suggestions. We have rewritten the policy recommendations based on the conclusions and moved the policy recommendations to the end in the ‘Manuscript’ (Lines 435-451).

Reviewer #2: 

1. the research contributions are not clear. A potential innovation in this manuscript is its consideration of using coupling model to analysis the relationship between environment, economy and tourism within China. previous papers have included this development and its application in understanding China’s coordination development between economy and tourism and environment. If there is no innovation in the method, what is your most important contribution? 

Response: Thanks for your suggestions. We are sorry that the innovations of the paper was not clear in the original manuscript. The innovations of the paper includes two parts:1)Previous studies mainly focused on single province in China, the studies based on large scale are lack; 2)there are fewer studies to analyze the spatial changes of China’s coordination development between environment and economy and tourism, and the paper makes up for this gap (Lines 72-85).

2. you can refer to some more professional research to standardize the expression, for instance, coupling coordination degree or coordination degree? Tourism is an integral part of the economy, thus, Is it feasible to use “coupling” to describe the relationship between tourism industry and economy. 

Response: Thanks for your suggestions. We refer to the references by Sheng et al.(2009), Yuan et al.(2014) and Zhou et al.(2016a, 2016b, 2016c), all of which use coupling coordination to study the relationship between tourism industry and the economy. We believe that it is feasible to use “coupling coordination” to describe the relationship between tourism industry and economy.

These references are as follows:

[1] Sheng Y C, Zhong Z P. Study on the Coupling Coordinative Degree between Tourism Industry and Regional Economy: A Case Study of Hunan Province. Tourism Tribune. 2009; 24(8): 23-29. 

[2] Yuan Y, Jin M, Ren J, Hu M, Ren P. The dynamic coordinated development of a regional environment-tourism-economy system: A case study from western Hunan province, China. Sustainability. 2014; 6(8): 5231-5251. https://doi.org/10.3390/su6085231

[3] Zhou C, Feng X G, Tang R. Analysis and forecast of coupling coordination development among the regional economy-ecological environment-tourism industry: A case study of provinces along the Yangtze economic zone. Economic Geography. 2016a; 36: 186-193. 

[4] Zhou C, Feng X G, Tang R. Analysis and forecast of coupling coordination development among the regional economy-ecological environment-tourism industry - A case study of provinces along the Yangtze Economic Zone. Economic Geography. 2016b; 36(3): 186–193.

[5] Zhou C, Jin C, Zhao B, Zhang F. The provincial difference of coupling coordinative development of regional economy-ecology-tourism. Journal of Arid Land Resources and Environment. 2016c; 30(7):203-208. 

3. the manuscript contains statements that are too hard to understand. It's almost like the author(s) had run out of steam at this stage and as the reader, I must say that I felt relieved to have completed the marathon! Specific comments with regard to grammar, flow and basic expression are too numerous to mention. Assistance with writing skills and grammar is needed through an independent third party - someone who knows little about the technical content but can assist with these shortcomings. The paper falls short of being a professionally written management article. 

Response: Thanks for your suggestions. This manuscript has been revised in the English format. Before we submitted the original manuscript, Dr. Yan Dan has helped us revise the English format. She received her doctorate degree from Wageningen University, and her English level reached the native language level. Before we submitted the manuscript, we employed American Journal Experts (AJE) to revise the English format. The editing certificate is as follows.

4. It needs a major rewrite and rethink to eliminate redundant material and unrelated debate and discussion. For example, the sentence in lines 28-30, page 1; the last sentence in 3.4, page 15, etc. 

Response: Thanks for your suggestions. We have rewritten the introduction in the ‘Manuscript’ (Lines 27-85). And we have rewritten the policy recommendations based on the conclusions and moved the policy recommendations to the end in the ‘Manuscript’ (Lines 435-451).

5.The results are not clearly presented, for example, especial spatial distribution pattern of coupling coordination development for the 31 provinces. 

Response: Thanks for your suggestions. We have converted The Table 2,3, and 4 to the Figure 2,3, and 4 in the ‘Manuscript’(Lines 286-288, 345-347, 401-402) and analyzed the spatial distribution pattern of coupling coordination development for the 31 provinces.

6.Some technical erros: “they have good representativeness, understanding, availability, integrity and dissemination”(lines 3-4, page 4), pay attention to “understanding” “are negative indexes (the larger the better), the other indexes are all positive indicators (the smaller the better)”((lines 3-4, page 4), for positive indicators, It should be “the larger the better”

Response: Thank you very much for reviewers’ suggestions. We have corrected the sentence in the ‘manuscript’ (Lines 97 & 98).

7.Section 3.4 contains statements that are just far too basic and obvious. Policy recommendations are not targeted, they should be given on the basis of your results specifically. 

Response: Thanks for your suggestions. We have rewritten the policy recommendations based on the conclusions and moved the policy recommendations to the end in the ‘Manuscript’ (Lines 435-451).

Reviewer #3: 

1.Line 7 of the first Paragraph in Section 1: “The impact and pressure of this extensive economic development has (should be have) led to…”. There exists an inappropriate tense.

Response: Thanks for your suggestions. We have rewritten the introduction in the ‘Manuscript’ (Lines 27-85).

2.Line 1 of the second Paragraph in Section 1: “…, much literature has studied the interaction …”. There exists an inappropriate tense.

Response: Thanks for your suggestions. We have corrected the sentence in the ‘manuscript’ (Lines 46-47).

3. Lines 1, 2 & 3 of the last Paragraph in Section 1: “…, whether the ecological environment can be coordinated with economic and tourism development has become a problem that many scholars and government departments need to study.”. The narrative here should be improved.

Response: Thanks for your suggestions. We have rewritten the introduction in the ‘Manuscript’ (Lines 27-85).

4. The first Section of the paper does not significantly explain what contributions it has made. It is suggested that the main contributions of the paper be listed in the last Paragraph of Section 1.

Response: Thanks for your suggestions. We are sorry that the main contributions of the paper have not been clearly expressed the main contributions in the original manuscript. The innovations of the paper includes two parts:1)Previous studies mainly focused on single province in China, the studies based on large scale are lack; 2)there are fewer studies to analyze the spatial changes of China’s coordination development between environment and economy and tourism, and the paper makes up for this gap (Lines 72-85).

5. The entropy weighting results of the three subsystems in Section 2.4 are not the focus of the paper. It is recommended to replace Table 1 with a concise language narrative.

Response: Thanks for your suggestions. We have revised the Section 2.4 and replaced Table 1 with a concise language narrative (Lines 205-207).

6. The results in Sections 3.2 and 3.3 are relatively thorough, but the results expressed in tabular form are worse than the graphical representation. It is recommended to replace Tables 2, 3 and 4 with the corresponding graphical representation.

Response: Thanks for your suggestions. We have converted The Table 2,3, and 4 to the Figure 2,3, and 4 in the ‘Manuscript’(Lines 286-288, 345-347, 401-402) and analyzed the spatial distribution pattern of coupling coordination development for the 31 provinces.

7. The conclusions of Section 4 are too much and too long. It is recommended to reduce the length and express the main conclusions of the paper in a clear and concise language.

Response: Thanks for your suggestions. We have revised the ‘Conclusion and policy recommendations’ in the ‘Manuscript’ (Lines 404-451). 

At last, we tried our best to improve the manuscript and made some changes in the manuscript. We really appreciate it that Reviewers help up improve the quality of the research, and hope that the correction will meet with approval. Once again, thank you very much for your comments and suggestions.

Yours Sincerely

Zhizhu Lai and Zheng Wang

---

## [Decision Letter · Decision Letter 2]

15 Jan 2020

Coupling coordination between environment, economy and tourism：A case study of China

PONE-D-19-27598R2

Dear Dr. wang,

We are pleased to inform you that your manuscript has been judged scientifically suitable for publication and will be formally accepted for publication once it complies with all outstanding technical requirements.

With kind regards,

Bing Xue, Ph.D.

Academic Editor

PLOS ONE

Additional Editor Comments (optional):

Reviewers' comments:

Reviewer's Responses to Questions

**Comments to the Author**

1. If the authors have adequately addressed your comments raised in a previous round of review and you feel that this manuscript is now acceptable for publication, you may indicate that here to bypass the “Comments to the Author” section, enter your conflict of interest statement in the “Confidential to Editor” section, and submit your "Accept" recommendation.

Reviewer #1: All comments have been addressed

Reviewer #3: All comments have been addressed

2. Is the manuscript technically sound, and do the data support the conclusions?

Reviewer #1: Yes

Reviewer #3: Yes

3. Has the statistical analysis been performed appropriately and rigorously? 

Reviewer #1: Yes

Reviewer #3: Yes

4. Have the authors made all data underlying the findings in their manuscript fully available?

Reviewer #1: Yes

Reviewer #3: Yes

5. Is the manuscript presented in an intelligible fashion and written in standard English?

Reviewer #1: Yes

Reviewer #3: Yes

6. Review Comments to the Author

Reviewer #1: In this paper, a comprehensive evaluation index system of the ecological environment, the economy and tourism is established, and the coupling degrees and coordination degrees of the three subsystems of 31 provinces in China from 2003 to 2017 are calculated. Overall, the authors have made further improvements accordingly, and I recommend its publication.

Reviewer #3: Thank you for the revisions on the manuscript. I appreciate the thoughtful responses to the reviewers’ comments. In particular, the replacement of tables with graphics makes the manuscript better, and the revised introduction and policy recommendations are more reasonable. Therefore, I suggest that this manuscript be accepted without further modification.

7. PLOS authors have the option to publish the peer review history of their article (what does this mean?). If published, this will include your full peer review and any attached files.

Reviewer #1: No

Reviewer #3: No

---

## [Editor Report · Acceptance letter]

23 Jan 2020

PONE-D-19-27598R2 

Coupling coordination between environment, economy and tourism: A case study of China 

Dear Dr. wang:

I am pleased to inform you that your manuscript has been deemed suitable for publication in PLOS ONE. Congratulations! Your manuscript is now with our production department. 

With kind regards,

on behalf of

Professor Bing Xue 

Academic Editor

PLOS ONE